**Data Availability Statement:** Anonymized datasets used and/or analysed during the current study are available at OpenBU, Boston University's open

# Integration of point-of-care screening for type 2 diabetes mellitus and hypertension with COVID-19 rapid antigen screening in Johannesburg, South Africa

Alana T. Brennan[1,2,3]*, Beatrice Vetter[4], Mohammed Majam[5], Vanessa T. Msolomba[5], Francois Venter[5], Sergio Carmona[4], Kekeletso Kao[4], Adena Gordon[1], Gesine Meyer-Rath[1,2,6]

1 Department of Global Health, Boston University School of Public Health, Boston, MA, United States of America, 2 Health Economics and Epidemiology Research Office, University of the Witwatersrand, Johannesburg, South Africa, 3 Department of Epidemiology, Boston University School of Public Health, Boston, MA, United States of America, 4 FIND, Geneva, Switzerland, 5 Ezintsha, Faculty of Health Sciences, University of the Witwatersrand, Johannesburg, South Africa, 6 South African DSI-NRF Centre of Excellence in Epidemiological Modelling and Analysis (SACEMA), Stellenbosch University, Stellenbosch, South Africa

* abrennan@bu.edu

## Abstract

### Aims

We sought to evaluate the yield and linkage-to-care for diabetes and hypertension screening alongside a study assessing the use of rapid antigen tests for COVID-19 in taxi ranks in Johannesburg, South Africa.

### Methods

Participants were recruited from Germiston taxi rank. We recorded results of blood glucose (BG), blood pressure (BP), waist circumference, smoking status, height, and weight. Participants who had elevated BG (fasting ≥7.0; random ≥11.1mmol/L) and/or BP (diastolic ≥90 and systolic ≥140mmHg) were referred to their clinic and phoned to confirm linkage.

### Results

1169 participants were enrolled and screened for elevated BG and elevated BP. Combining participants with a previous diagnosis of diabetes (n = 23, 2.0%; 95% CI:1.3–2.9%) and those that had an elevated BG measurement (n = 60, 5.2%; 95% CI:4.1–6.6%) at study enrollment, we estimated an overall indicative prevalence of diabetes of 7.1% (95% CI:5.7–8.7%). When combining those with known hypertension at study enrollment (n = 124, 10.6%; 95% CI:8.9–12.5%) and those with elevated BP (n = 202; 17.3%; 95% CI:15.2–19.5%), we get an overall prevalence of hypertension of 27.9% (95% CI:25.4–30.1%). Only 30.0% of those with elevated BG and 16.3% of those with elevated BP linked-to-care.

access repository (https://hdl.handle.net/2144/46066).

**Funding:** ATB, MM, VTM, FV, AG and GMR received funding from FIND through a grant from the German Federal Ministry of Economic Cooperation and Development. ATB received funding from the National Institute of Diabetes and Digestive and Kidney Diseases (NIDDK) grant 1K01DK116929-01A1.

**Competing interests:** No authors have competing interests. BV, SC and KK are employees of FIND.

## Conclusion

By opportunistically leveraging existing COVID-19 screening in South Africa to screen for diabetes and hypertension, 22% of participants received a potential new diagnosis. We had poor linkage-to-care following screening. Future research should evaluate options for improving linkage-to-care, and evaluate the large-scale feasibility of this simple screening tool.

## Introduction

Prior to the COVID-19 pandemic, South Africa had entered a health transition in which mortality rates from non-communicable chronic diseases (NCDs) started to surpass mortality rates from infectious diseases [1], with type 2 diabetes mellitus (diabetes) having an estimated adult prevalence of 11% and hypertension at 30% [2, 3]. Challenges like suboptimal screening, diagnosis and management of diabetes and hypertension coupled with poor access to diagnostic tests remains frequent in clinical practice in low- and middle-income countries (LMICs) [4, 5]. As such, improved access to diagnostic tests is an integral component of optimal management of diabetes and hypertension and is necessary for reducing NCD-related morbidity and mortality.

Although possibly transitory in nature, COVID-19 derailed many gains in LMICs in screening, diagnosis and management of NCDs. According to a recent World Health Organization survey of 155 countries, 95% reported employees working in NCD care being reassigned to support COVID-19 efforts, while more than 50% of countries reported postponing public health screening programs for diabetes and hypertension [6]. This impact is of significant concern given that people living with NCDs, specifically diabetes and hypertension, are in turn at higher risk of severe COVID-19-related illness and death [7]. Maintaining screening, diagnosis, quality care and treatment of people with these conditions is critical.

In South Africa, screening for diabetes and hypertension is predominantly done in a primary health care (PHC) setting. The government provides algorithms for health personnel to provide care in a stepwise fashion for both conditions [8]. However, South Africa's PHC clinics have been facing an unmanageable workload for the decades [9], so adequate screening, diagnosis and care and treatment for people living with diabetes and/or hypertension in this setting is difficult.

The large proportion of the population that has elevated blood glucose and/or blood pressure that are either unaware of their condition or are aware and uncontrolled [10] are partly a results of a poorly managed PHC system. As such, simple and effective community-based routine screening outside of the PHC setting for NCDs can help strengthen local healthcare systems, ease the burden on providers, increase access to healthcare, and improve patient outcomes, by diagnosing individuals quickly and earlier, shortening the time to seek treatment for these conditions [11, 12]. Community-level screening for tuberculosis and HIV has been integral components of standard testing algorithms in sub-Saharan Africa since 2004, but there continues to be a lack of use of screening for NCDs [13, 14].

COVID-19 community screening programs provide temporary opportunities where individuals are already gathered to seek medical care. The objective of our study was therefore to evaluate the feasibility, yield, and linkage-to-care of rapid screening for diabetes and hypertension alongside the existing COVID-19 digital health supported antigen rapid diagnostic test (Ag-RDT) field study [15].

## Methods

This prospective cohort study was carried out from 3 August-29 September 2021, embedded into an operational study investigating the use of Ag-RDTs for COVID-19 in busy public transport hubs (taxi ranks) in Johannesburg, South Africa [15]. Participants for our study were recruited from the Germiston taxi rank. After participants were screened for study eligibility, consented, enrolled and had completed the COVID-19 risk score screening, study staff conducted a random glucose test (via glucose meter), and measured blood pressure (taken while the individual was seated via a once-off measurement with a manual blood pressure device with cuff), waist circumference in centimeters (via measuring tape), height in centimeters, and weight in kilograms (via scales and a height measuring rod). All study relevant measures were recorded before nasopharyngeal swabs for COVID-19 testing were administered, eliminating the role these swabs could have had on elevating blood pressure. Fasting was defined as participants who had nothing but water in the previous 8 hours before study enrollment (n = 286, or 24.5%). Participants who had elevated blood glucose (random glucose $\geq$11.1 mmol/L; fasting glucose $\geq$7.0 mmol/L [8, 16]) and/or blood pressure (diastolic $\geq$90 mmHg; systolic $\geq$140 mmHg [8, 17]) were provided with a written referral indicating the results of blood glucose and blood pressure screening and were encouraged to link to their local PHC for confirmation and management of their diabetes and/or hypertension. A formal laboratory test (glycated hemoglobin A1c (HbA1c), fasting, or random blood glucose) is required to confirm diagnosis, while repeat testing is required to confirm a hypertension diagnosis [8]. Participants referred to their local PHC were contacted once a week for three weeks to confirm if they had visited their PHC for the necessary follow-up.

### Informed consent

The principles if informed consent in the current edition of the Declaration of Helsinki were implemented before any protocol-specified procedures or interventions were carried out. The consent form described the purpose of the study, the procedures to be followed, and the risks and benefits of participation. Potential participants had the opportunity to have any questions answered before and after completing the screening questionnaire or the informed consent form. Participants signed the informed consent form electronically (paper based informed consent forms were used as back-ups when the electronic system was down), before any project procedures were performed. For this study we explained the informed consent in the participants preferred language but informed the participant the electronic participant declaration would be in English. Participants were able to request a copy of this to keep digitally for their own records. Participants were told that they have the right to withdraw from the study at any time. If a participant withdrew from the study, their research data was removed from all platforms and was not included in the analysis.

### Ethics statement

Approval for analysis of the de-identified cohort was granted by Boston University's Institutional Review Board (Protocol No. H-42030), Human Research Ethics Committee of the University of the Witwatersrand (Protocol No. M210411).

### Study population

Inclusion and exclusion criteria for our pilot study followed the same criteria for the COVID-19 digital health supported Ag-RDT field study [15]. As such, all commuters, vendors, and drivers at the Germiston taxi rank who were $\geq$18 years old and literate were invited to

participate. Participants had to have a mobile phone capable of receiving unstructured supplementary service data, SMS, or WhatsApp messaging. The study excluded 1) participants who refused consent or were unable to provide informed consent; 2) any participants with contraindications to nasopharyngeal sample collection for the COVID-19 Ag-RDT; 3) vulnerable populations as deemed inappropriate for the study by study personnel; 4) personnel directly involved in the conduct of the study; 5) participants at risk of failing to comply with the provisions of the protocol as to cause harm to self or seriously interfere with the validity of the study results; and 6) participants who had a confirmed positive COVID-19 diagnosis up to 3 months prior.

## Outcomes

Our primary outcomes of interest were as follows:

1. number of participants with elevated blood glucose level (i.e., $\geq$11.1 mmol/L if measurement is random; $\geq$7.0 mmol/L if measurement is equivalent to fasting)

2. number of participants with elevated blood pressure (i.e. diastolic $\geq$90 mmHg; systolic $\geq$140 mmHg)

3. number of participants classified as pre-obese (body mass index (BMI) 25.0–29.9 kg/m$^2$), obese (BMI 30.0–39.9 kg/m$^2$) and severely obese (BMI $> = 40$ kg/m$^2$)

4. number of male participants with a waist circumference >90 cm [18] and female participants >91.5 cm [19], indicating risk of metabolic syndrome

5. number of participants with both elevated GL and BP in addition to a waist circumference indicative of metabolic syndrome and/or classified as pre-obese/obese/severely obese

6. number of participants in outcomes 1, 2, and 5 who when contacted stated that they linked to primary healthcare for further diabetes and/or hypertension testing and care.

## Statistical analysis

We used descriptive statistics to display the clinical and demographic characteristics of our study population. For our primary outcomes, we used descriptive statistics stratified by biological sex and crude and adjusted modified Poisson regression to assess predictors of elevated blood glucose, elevated blood pressure, and linkage-to-care. We modeled the outcome of elevated blood glucose as a function of age (18–29.9, 30–39.9, 40–49.9, 50–59.9, $\geq$60 years), sex, BMI (categorized as <25 vs. $\geq$25 kg/m$^2$), smoking status (ever vs. never), elevated blood pressure at enrollment (diastolic $\geq$90 mmHg and systolic $\geq$140 mmHg), previous diabetes diagnosis at enrollment (self-reported) and previous hypertension diagnosis at enrollment (self-reported). The outcome of elevated blood pressure was adjusted for the same variables, with the exception of elevated blood pressure at enrollment being replaced by elevated blood glucose at enrollment. The outcome of linkage-to-care was modeled as a function of all covariates mentioned. All analyses were conducted using SAS v. 9.4.

## Results

### Cohort

A total of 1169 participants were enrolled during an eight-week recruitment period and screened for elevated blood glucose and elevated blood pressure. The median age of participants was 37.0 years [interquartile range (IQR):30.0–47.0] and 58.3% were men (Table 1). The

**Table 1. Characteristics and demographics of participants screened for elevated blood pressure and blood glucose at Germiston, South Africa taxi rank (N = 1168).**

| | Male | Female | Total |
|---|---|---|---|
| | n = 682 (58.3) | n = 486 (41.6) | N = 1168* |
| **Age (years)** (n,%) | | | |
| 18–29 | 146 (21.4) | 114 (23.5) | 260 (22.2) |
| 30–39 | 218 (32.0) | 167 (34.4) | 386 (33.0) |
| 40–49 | 180 (26.4) | 115 (23.7) | 295 (25.2) |
| 50–59 | 106 (15.5) | 61 (12.6) | 167 (14.3) |
| ≥60 | 32 (4.7) | 29 (6.0) | 61 (5.2) |
| **Age (median; IQR)** | 38.0 (31.0–48.0) | 36.0 (30.0–47.0) | 37.0 (30.0–47.0) |
| **COVID Status** (n,%) | | | |
| Positive | 3 (0.4) | 8 (1.7) | 11 (0.9) |
| **BMI categories** (n,%) | | | |
| underweight ($<18.5$ kg/m$^2$) | 36 (5.3) | 6 (1.2) | 42 (3.6) |
| normal (18.5–24.9 kg/m$^2$) | 326 (47.8) | 104 (21.4) | 430 (36.8) |
| pre-obese (25.0–29.9 kg/m$^2$) | 199 (29.2) | 137 (28.2) | 337 (28.8) |
| obese (30.0–39.9 kg/m$^2$) | 111 (16.3) | 203 (41.8) | 314 (26.9) |
| severely Obese ($> = 40$ kg/m$^2$) | 8 (1.2) | 35 (7.2) | 43 (3.7) |
| Missing | 2 (0.3) | 1 (0.2) | 3 (0.3) |
| **Body Mass Index (median; IQR)** | 24.4 (21.6–28.4) | 29.7 (25.4–33.8) | 26.5 (22.8, 31.1) |
| **Employment** (n,%) | | | |
| full-time | 406 (59.5) | 229 (47.1) | 636 (54.4) |
| part-time | 65 (9.5) | 53 (10.9) | 118 (10.1) |
| student | 13 (1.9) | 22 (4.5) | 35 (3.0) |
| Unemployed | 198 (29.0) | 182 (37.4) | 380 (32.5) |
| **Smoking Status** (n,%) | | | |
| ever | 214 (31.4) | 23 (4.7) | 237 (20.3) |
| never | 464 (68.0) | 459 (94.4) | 924 (70.0) |
| Missing | 4 (0.6) | 4 (0.8) | 8 (0.7) |
| **Patient Type** (n,%) | | | |
| commuter | 591 (86.7) | 437 (89.9) | 1029 (88.0) |
| driver | 52 (7.6) | 0 (0.0) | 52 (4.5) |
| vendor | 28 (4.1) | 15 (3.1) | 62 (5.3) |
| other | 11 (1.6) | 34 (7.0) | 26 (2.2) |
| **Previous diabetes diagnosis at enrollment** | 6 (0.9) | 17 (3.5) | 23 (2.0) |
| **Previous hypertension diagnosis at enrollment** | 49 (7.2) | 75 (15.4) | 124 (10.6) |
| **Outcomes (n (%; 95% CI)** | | | |
| elevated blood glucose [1] | 36 (5.3; 3.8–7.2) | 28 (5.8; 3.9–8.1) | 64 (5.5; 4.3–6.9) |
| elevated blood pressure [2] | 147 (21.6; 18.6–24.8) | 87 (17.9; 14.6–21.5) | 234 (20.0; 17.8–22.4) |
| elevated blood glucose no previous diabetes diagnosis [3] | 35 (5.2; 3.7–7.0) | 25 (5.3; 3.6–7.7) | 60 (5.2; 4.1–6.6) |
| elevated blood pressure no previous hypertension diagnosis [4] | 134 (21.2; 18.1–24.5) | 68 (16.5; 13.2–20.4) | 202 (19.3; 17.0–21.8) |
| classified as pre-obese, obese, and severely obese | 318 (46.6; 42.9–50.4) | 375 (77.3; 73.3–80.7) | 694 (59.4; 53.4–62.0) |
| waist circumference indicative of metabolic syndrome [5] | 4 (0.6; 0.2–1.4) | 4 (0.8; 0.3–2.0) | 8 (0.7; 0.3–1.3) |
| ≥3 or more risk factors above | 9 (1.3; 0.6–2.4) | 5 (1.0; 0.4–2.3) | 14 (1.2; 0.7–2.0) |
| Linkage-to-care amongst those with elevated blood glucose [6] | 13 (37.1; 22.5–53.9) | 5 (20.8; 8.1–40.3) | 18 (30.0; 19.4–42.4) |

*(Continued)*

**Table 1.** (Continued)

| | Male | Female | Total |
|---|---|---|---|
| | **n = 682 (58.3)** | **n = 486 (41.6)** | **N = 1168\*** |
| Linkage-to-care amongst those with elevated blood pressure [7] | 20 (14.9; 9.6–21.7) | 13 (19.1; 11.1–29.8) | 33 (16.3; 11.7–21.9) |

\*one participant is missing gender

[1] ≥11.1 mmol/L if measurement is random; ≥7.0 mmol/L if measurement is equivalent to fasting

[2] diastolic ≥90 mmHg and systolic ≥140 mmHg

[3] denominator male n = 676 and female n = 469

[4] denominator male n = 633 and female n = 411

[5] >90 cm (males), >91.5 cm (females)

[6] denominator male n = 35 and female n = 25

[7] denominator male n = 134 and female n = 68

median BMI for all participants was 26.5 kg/m$^2$ [IQR:22.8–31.1], 28.8% were identified as pre-obese, 26.9% as obese, and 3.7% as severely obese. There was a higher prevalence of obesity (41.8% vs. 16.3%, respectively) and severe obesity (7.2% vs. 1.2%, respectively) among females compared to males. A total of 237 (20.3%) of participants identified as ever having smoked, with the prevalence of smoking being higher in males (31.4%) compared to females (4.7%). The majority of participants were employed full-time (54.4%), which was also higher in males (59.5%) than in females (47.1%). The COVID-19 positivity rate was low at 0.9% (n = 11), and higher in females (1.7%) compared to males (0.4%).

## Primary outcomes

Combining participants with a previous diagnosis of diabetes (n = 23, 2.0%; 95% confidence interval (CI):1.3–2.9%) and those that had an elevated blood glucose measurement (n = 60, 5.2%; 95% CI:4.1–6.6%) at study enrollment, we estimated an overall indicative prevalence of diabetes of 7.1% (95% CI:5.7–8.7%) (Table 1 and Fig 1). The 60 participants or 72% of the total 83 participants unaware of their elevated blood glucose were predominately male (58.3%, 95% CI:45.6–70.3%). We found that women (3.5%; 95% CI:2.1–5.4%) had a higher prevalence of a previous diabetes diagnosis at study enrollment than their male counterparts (0.9%; 95% CI:0.4–1.8%). However, we saw no difference in the prevalence of elevated blood glucose levels by sex (female 5.8%; 95% CI:3.9–8.1% vs. male 5.3%; 95% CI:3.8–7.2%) amongst those with no known diabetes diagnosis at enrollment. A total of 4 (17.4%; 95% CI:5.8–36.8%) participants that had a previous diagnosis of diabetes had elevated blood sugar levels at enrollment, 3 (75.0%) of which were female.

When combining those with known hypertension at study enrollment (n = 124, 10.6%; 95% CI:8.9–12.5%) and those with elevated blood pressure (n = 202; 17.3%; 95% CI:15.2–19.5%), we get an overall prevalence of 27.9% (n = 326; 95% CI:25.4–30.1%) (Table 1 and Fig 2). As with elevated blood glucose, the majority of the 202 participants (or 62% of the total 326 participants) unaware of their elevated blood pressure were male (66.3%, 95% CI:59.6–72.6%). We found that women (15.4%; 95% CI:12.4–18.9%) had a higher prevalence of hypertension diagnosis at study enrollment than their male counterparts (7.2%; 95% CI:5.4–9.3%). However, we saw a more comparable relationship when stratifying elevated blood pressure levels by sex (female 17.9%; 95% CI:14.6–21.5% vs. male 21.6%; 95% CI:18.6–24.8%) amongst those with no known hypertension diagnosis. A total of 32 (25.8%; 95% CI:18.7–34.0%) participants who had a previous diagnosis of hypertension had uncontrolled blood pressure at enrollment, 59.4% (n = 19) of whom were female.

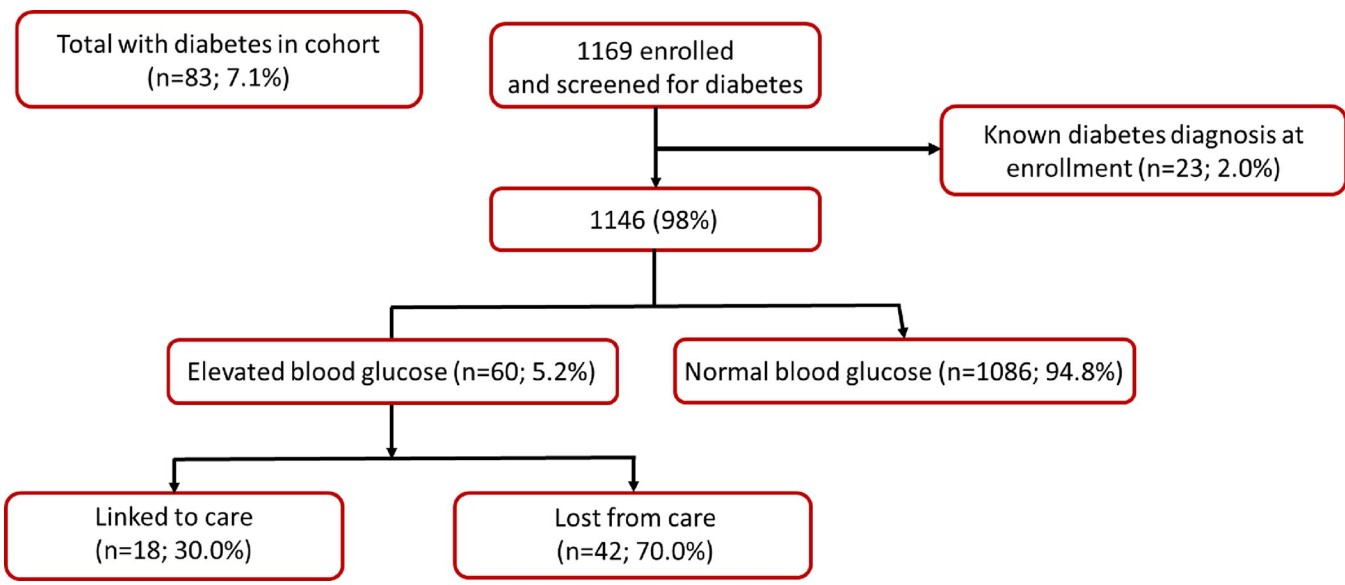

Fig 1. Number of clients with elevated blood glucose level indicative of diabetes mellitus (i.e., ≥11.1 mmol/L if measurement is random; ≥7.0 mmol/L if measurement is equivalent to fasting) amongst those with no known previous diabetes diagnosis at study enrollment (N = 1169).

Having a waist circumference indicative of metabolic syndrome (>90 cm for males and >91.5 cm for females) was rare in our cohort (0.7%; 95% CI:0.3–1.3%) (Table 1), while the number of participants with elevated blood glucose and blood pressure in addition to a high waist circumference and/or classified as pre-obese/obese/severely obese was also rare at 1.2% (95% CI:0.7–2.0%).

Amongst those participants with elevated blood glucose that had no known previous diagnosis of diabetes (n = 60), 18 (n = 30.0%; 95% CI:19.4–42.4%) linked to care within three weeks of study enrollment (Table 1 and Fig 1), while amongst those with elevated blood

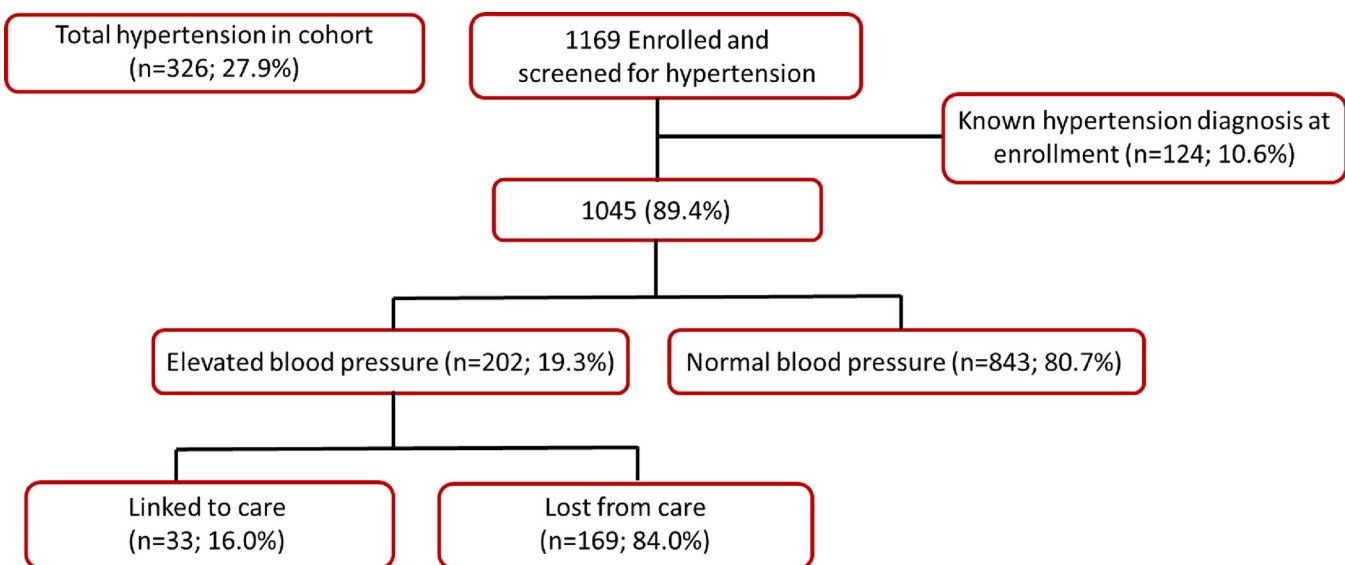

Fig 2. Number of clients with elevated blood pressure indicative of hypertension (i.e. diastolic ≥90 mmHg; systolic ≥140 mmHg) amongst those with no known previous diagnosis of hypertensive at study enrollment (N = 1169).

pressure and no known hypertension diagnosis (n = 202), 33 (16.3%; 95% CI:11.7–21.9%) linked to care (Table 1 and Fig 2). Of those participants with elevated blood glucose, more males (n = 13, 37.1%; 95% CI:22.5–53.9%) sought out PHC services than females (n = 5; 20.8%; 95% CI:8.1–40.3%), but more females (n = 13; 19.1%; 95% CI:11.1–29.8%) with elevated blood pressure linked to PHC services than males (n = 20; 14.9%; 95% CI:9.6–21.7%) (Table 1).

## Predictors of elevated blood glucose, elevated blood pressure and linkage-to-care

The modified Poisson regression model we used to assess predictors of elevated blood glucose suggests that those ≥30 years of age, individuals with elevated blood pressure at enrollment (adjusted risk ratio (aRR) 1.96; 95% CI:1.15–3.33) and those with a BMI ≥25kg/m$^2$ vs. <25kg/m$^2$ (aRR 2.05; 95% CI:1.06–3.95) were at increased risk of elevated blood glucose (Table 2). Our results, although imprecise, also suggest participants with a previous diagnosis of hypertension (aRR 1.54; 95% CI:0.81–2.95), those with a previous diagnosis of diabetes (aRR 2.24; 95% CI:0.77–6.57), and those that had ever smoked vs never smokers (aRR 1.59; 95% CI:0.85, 2.97) at enrollment were also at increased risk of elevated blood glucose.

**Table 2. Crude and adjusted risk ratios assessing predictors of elevated blood glucose and blood pressure (n = 1169).**

|  | elevated blood glucose | | elevated blood pressure | |
|---|---|---|---|---|
|  | RR (95% CI) | aRR (95% CI) | RR (95% CI). | aRR (95% CI) |
| **Age (years)** |  |  |  |  |
| 18–29 | ref | ref | ref | ref |
| 30–39 | 1.97 (0.78–4.96) | 1.42 (0.55–3.65) | 2.46 (1.44–4.20) | 2.06 (1.20–3.54) |
| 40–49 | 2.56 (1.02–6.46) | 1.69 (0.65–4.39) | 4.51 (2.68–7.58) | 3.50 (2.07–5.92) |
| 50–59 | 4.05 (1.58–10.34) | 2.38 (0.89–6.35) | 4.49 (2.58–7.79) | 3.24 (1.84–5.70) |
| ≥60 | 4.13 (1.33–12.81) | 2.03 (0.60–6.88) | 4.76 (2.48–9.17) | 3.44 (1.74–6.79) |
| **Gender** |  |  |  |  |
| Male | ref | ref | ref | Ref |
| Female | 1.09 (0.66–1.78) | 1.07 (0.61–1.89) | 0.83 (0.64–1.08) | 0.66 (0.50–0.88) |
| **Previous hypertension diagnosis** |  |  |  |  |
| No | ref | Ref | ref | Ref |
| yes | 2.14 (1.17–3.94) | 1.54 (0.81–2.95) | 1.34 (0.92–1.94) | 1.09 (0.74–1.61) |
| **Previous diabetes diagnosis** |  |  |  |  |
| No | ref | ref | ref | Ref |
| yes | 3.29 (1.20–9.05) | 2.24 (0.77–6.57) | 1.09 (0.45–2.64) | 0.72 (0.28–1.81) |
| **Elevated blood pressure at enrollment** |  |  |  |  |
| No | ref | ref | – | – |
| yes | 2.53 (1.53–4.18) | 1.96 (1.15–3.33) | – | – |
| **Elevated blood glucose diagnosis at enrollment** |  |  |  |  |
| No | – | – | ref | Ref |
| yes | – | – | 2.04 (1.35–3.10) | 1.60 (1.05–2.44) |
| **Body Mass Index** |  |  |  |  |
| <25.0 kg/m$^2$ | ref | ref | ref | Ref |
| ≥25.0 kg/m$^2$ | 2.68 (1.46–4.93) | 2.05 (1.06–3.95) | 2.43 (1.78–3.31) | 2.23 (1.61–3.08) |
| **Smoking Status** |  |  |  |  |
| never | ref | ref | ref | ref |
| ever (current or former) | 1.31 (0.74–2.31) | 1.59 (0.85–2.97) | 0.80 (0.57–1.13) | 0.77 (0.53–1.10) |

**Table 3. Predictors of linkage-to-care amongst those with elevated blood glucose and/or blood pressure (n = 234).**

| | Linked to Care | |
| --- | --- | --- |
| | RR (95% CI) | aRR* (95% CI) |
| **Age** | | |
| <40 | ref | ref |
| ≥40 | 2.28 (1.14–4.53) | 2.01 (0.99–4.11) |
| **Comorbidities** | | |
| elevated blood glucose | ref | Ref |
| elevated blood pressure | 0.83 (0.38–1.78) | 0.68 (0.31–1.51) |
| both | 1.87 (0.75–4.65) | 1.23 (0.46–3.29) |
| **Previous hypertension diagnosis** | | |
| No | ref | Ref |
| yes | 1.99 (1.05–3.79) | 1.79 (0.91–3.49) |
| **Previous diabetes diagnosis** | | |
| No | ref | Ref |
| yes | 1.49 (0.36–6.12) | 0.86 (0.20–3.71) |
| **Gender** | | |
| Male | ref | Ref |
| female | 1.25 (0.73–2.15) | 1.00 (0.55–1.82) |
| **BMI** | | |
| <25.0 kg/m$^2$ | ref | Ref |
| ≥25.0 kg/m$^2$ | 2.35 (1.00–5.50) | 2.06 (0.86–4.93) |
| **Smoking Status** | | |
| never | ref | Ref |
| ever (current or former) | 0.89 (0.42–1.89) | 0.78 (0.34–1.67) |

*age was collapsed to <40 vs. ≥40 years due to smaller sample size

Our models used to assess predictors of elevated blood glucose show that those ≥30 vs. <30 years of age, participants with elevated blood glucose (aRR 1.60; 95% CI:1.05–2.44) and individuals with a BMI ≥25kg/m$^2$ vs. <25kg/m$^2$ (aRR 2.23; 95% CI:1.61–3.08) at enrollment were at increased risk of elevated blood pressure levels (Table 2). We also found that females had a 34% decreased risk (aRR 0.66; 95% CI 0.50–0.88) of elevated blood pressure levels when compared to their male counterparts.

For the outcome of linkage-to-care, our results, although imprecise most likely due to small sample size, suggest that those ≥40 years of age, participants with a previous diagnosis of hypertension (aRR 1.79; 95% CI:0.91–3.49) and individuals with a BMI ≥25kg/m$^2$ vs. <25kg/m$^2$ (aRR 2.06; 95% CI:0.86–4.93) were more likely to linkage-to-care (Table 3). We also found, although imprecise, that those with only elevated blood pressure at enrollment were less likely to seek care than those with only elevated blood sugar (aRR 0.68; 95% CI:0.31–1.51), while those with both elevated blood glucose and blood pressure were more likely to seek care compared to those with only elevated blood glucose (aRR 1.23; 95% CI:0.46–3.29).

## Discussion

Recognition and prevalence of NCDs have risen throughout sub-Saharan Africa. Many NCDs can be prevented or treated early on with low-cost interventions, yet implementation of such care has been limited throughout the region due to an already overburdened health care system, further impacted by the COVID-19 pandemic. Our study provides evidence of the

feasibility of leveraging existing COVID-19 health screening infrastructure in South Africa to provide simple screening for elevated blood pressure and elevated blood glucose outside of health facilities at a high traffic taxi rank in Germiston. While community COVID-19 screening efforts might fluctuate or eventually stop in the months to come, the screening intervention could be easily added into other health staff-led mobile testing effort, such as those for HIV and tuberculosis.

We screened 1169 participants during an eight-week period and found an overall indicative prevalence of diabetes of 7.1% (combining previous diagnosis of diabetes (2%) and those that had an elevated blood glucose measurement (5.2%) at study enrollment. Our estimate was slightly lower than what has been previously reported for South Africa (11% [2, 20]), but in line with more recent estimates from sub-Saharan Africa (7.2%) [21]. We found females had a higher prevalence of previously diagnosed diabetes (3.5%) compared to males (0.9%) at enrollment. Research suggests that this may be attributable to men having a higher risk of dying from other causes (e.g., HIV, tuberculosis, accidents) prior to being diagnosed with diabetes, or due to differences in health seeking behavior by sex [22]. Once we removed participants with known diabetes at enrollment, we found estimates of elevated blood glucose to be comparable between men and women in our study, which is consistent with previous research [23].

Our overall hypertension prevalence was 27.9% (combining those with known hypertension at study enrollment (10.6%) and those with elevated blood pressure (17.3%)), which is comparable to those previously reported out of South Africa [24–26]. We found that women (15.4%) had a higher prevalence of previous diagnosed hypertension at study enrollment than men (7.2%), also consistent with previous work [24–28]. These differences by sex in the prevalence of hypertension, like diabetes, could be associated with higher magnitudes of obesity and physical inactivity among women in our study population [24–28]. However, we saw the relationship become more comparable (overlapping confidence intervals in our estimates) when stratifying elevated blood pressure levels by sex (female 16.5% vs. male 21.2%) amongst those with no known hypertension diagnosis, consistent with previous research [29].

One of the most interesting findings of our study were the additional 60 (72% of the total 83 individuals potentially living with diabetes) and additional 202 (62% of the total 326 individuals potentially living with hypertension) participants in our cohort being newly diagnosed with either condition, equating to almost one quarter of screened participants being unaware of their elevated blood glucose or blood pressure levels. This observation is common in many studies of diabetes [23, 30] and hypertension [28, 31] in sub-Saharan Africa. With more than 50% of people living with diabetes unaware of their condition [23, 32, 33], and 7% to 56% of people living with hypertension being unaware of their blood pressure status [25, 34]. We found that those unaware of their blood glucose or blood pressure levels were predominately male, possibly attributable to men generally being less engaged in the health system than women [23].

We found 17.4% of those with a prior diabetes diagnosis and 26.0% of those with a prior diagnosis of hypertension at enrollment did not have their condition under control, the majority of which were female. Our estimates are lower than what has been reported previously in sub-Saharan Africa (uncontrolled diabetes ranging from 18.1% to 30.3% [20, 30] and uncontrolled hypertension ranging from 50% to 93% [32–34]). The difference between our estimates and those previously reported could be that most of the previous estimates are from household surveys with a denominator comprised of everyone who was found in testing as part of the survey, while our denominator was comprised of individuals of people who knew they had hypertension and/or diabetes. It also could be that disease control once someone is engaged with the health system is higher in this setting, or those who self-report a prior diagnosis are more likely to be engaged in care and adherent to treatment, even in the context of the COVID-19 pandemic.

Linkage-to-care was poor in our study. It could be that moving testing for diabetes and hypertension into an individual's commute makes it more difficult to link-to-care compared to being screened and diagnosed at a health facility where an individual could be enrolled in care for their condition and initiated onto treatment immediately, at the same clinic. A recent study assessing the intermediary cost-effectiveness of distributing HIV self-test kits and onward linkage to confirmatory testing and treatment services through 11 distribution models in South Africa reported that moving the location of HIV testing from the facility to the community helped close the gap in knowledge of HIV status but increased the gap in the next step, from diagnosis to engaging in care, as the health facility was still far away from the community and severe barriers to care continued to exist [35]. Additionally, available evidence suggests that the COVID-19 pandemic severely impacted control and prevention programs for other conditions in South Africa, including diabetes and hypertension [36, 37], due to higher demand for COVID-19 related health services, process changes including. screening of patients at the gate, longer than usual waiting times and line cutoffs, the need to book a visit in advance) and overall hesitancy of individuals to visit health facilities for any services during the pandemic.

Additionally, although our sample size of patients referred to care after an elevated blood glucose and/or blood pressure measurement was small (n = 234) and we cannot draw strong conclusions, the predictors of linkage-to-care that we identified were comparable to what has been previously reported for chronic health conditions in this setting [20, 30]. Those ≥40 years, participants with a previous hypertension diagnosis at enrollment and those with a BMI ≥25 kg/m$^2$ were more likely to seek further care for their condition in our study.

There are many strengths of our study. We provide evidence that any engagement with the health system, however incidental, can be used as an opportunity to screen for additional diseases or co-morbidities. Additionally, our results might be generalizable to the general population in Southern African and possibly more broadly into other regions of sub-Saharan Africa that have high prevalence of obesity. It is important to consider our study alongside its limitations. First, a single measurement of blood glucose of blood pressure cannot be used to diagnose diabetes or hypertension. In order to get a definitive diagnosis, further formal laboratory measurements and clinical evaluations would be necessary, and in order to receive a full diagnosis and/or treatment the individual must link to the health care system [9, 16, 17]. Second, prior diagnoses of diabetes and hypertension were self-reported by participants; therefore, response bias is possible. Finally, there may be a recruitment bias, as taxi ranks generally service employed people travelling to or from work, as evidenced by the vast majority (64.5%) of participants in our study who were employed full- or part-time. Likely as a result of this, we enrolled more males (58.3%) than females into our study, which is reflective of men more likely to be employed than women in South Africa [38].

## Conclusion

In our proof of concept study, opportunistically leveraging the existing intervention of COVID-19 screening in South Africa to provide screening for hypertension and diabetes, we detected elevated blood glucose in 5.1% and elevated blood pressure in 17.3% of our participants with no known previous diagnosis of either condition, equating to almost 25% of screened participants being unaware of their elevated blood glucose or blood pressure levels. Though we identified these cases, there was poor linkage-to-care following the screening. Future research should evaluate options such as digital technologies to facilitating screening and linkage-to-care for diagnoses made outside of health facilities, and evaluate the feasibility and cost-effectiveness of this type of screening which can easily be extended to other community screening efforts beyond COVID-19- such as those for HIV and tuberculosis.

## Author Contributions

**Conceptualization:** Alana T. Brennan, Beatrice Vetter, Sergio Carmona, Gesine Meyer-Rath.

**Data curation:** Alana T. Brennan.

**Formal analysis:** Alana T. Brennan, Adena Gordon.

**Funding acquisition:** Alana T. Brennan, Gesine Meyer-Rath.

**Investigation:** Gesine Meyer-Rath.

**Methodology:** Gesine Meyer-Rath.

**Project administration:** Mohammed Majam, Vanessa T. Msolomba, Francois Venter, Kekeletso Kao.

**Resources:** Mohammed Majam, Vanessa T. Msolomba, Kekeletso Kao.

**Supervision:** Beatrice Vetter, Francois Venter, Sergio Carmona.

**Writing – original draft:** Alana T. Brennan, Beatrice Vetter, Francois Venter, Adena Gordon, Gesine Meyer-Rath.

**Writing – review & editing:** Alana T. Brennan, Beatrice Vetter, Mohammed Majam, Vanessa T. Msolomba, Francois Venter, Sergio Carmona, Kekeletso Kao, Gesine Meyer-Rath.

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
