## [Decision Letter · Decision Letter 0]

23 Mar 2023

PONE-D-22-33612Integration of point-of-care screening for type 2 diabetes mellitus and hypertension with COVID-19 rapid antigen screening in Johannesburg, South AfricaPLOS ONE

Dear Dr. Brennan,

Thank you for submitting your manuscript to PLOS ONE. After careful consideration, we feel that it has merit but does not fully meet PLOS ONE’s publication criteria as it currently stands. Therefore, we invite you to submit a revised version of the manuscript that addresses the points raised during the review process.

We look forward to receiving your revised manuscript.

Kind regards,

Mobolanle Balogun

Academic Editor

PLOS ONE

Journal Requirements:

2. Please include a complete copy of PLOS’ questionnaire on inclusivity in global research in your revised manuscript. Our policy for research in this area aims to improve transparency in the reporting of research performed outside of researchers’ own country or community. The policy applies to researchers who have travelled to a different country to conduct research, research with Indigenous populations or their lands, and research on cultural artefacts. The questionnaire can also be requested at the journal’s discretion for any other submissions, even if these conditions are not met.  Please find more information on the policy and a link to download a blank copy of the questionnaire here: https://journals.plos.org/plosone/s/best-practices-in-research-reporting. Please upload a completed version of your questionnaire as Supporting Information when you resubmit your manuscript

"ATB, MM, VTM, FV, AG and GMR received funding from The Foundation for Innovative New Diagnostics (FIND). "      

"No authors have competing interests"

Reviewers' comments:

Reviewer's Responses to Questions

**Comments to the Author**

1. Is the manuscript technically sound, and do the data support the conclusions?

Reviewer #1: Partly

Reviewer #2: Yes

2. Has the statistical analysis been performed appropriately and rigorously? 

Reviewer #1: No

Reviewer #2: Yes

3. Have the authors made all data underlying the findings in their manuscript fully available?

Reviewer #1: Yes

Reviewer #2: Yes

4. Is the manuscript presented in an intelligible fashion and written in standard English?

Reviewer #1: Yes

Reviewer #2: Yes

5. Review Comments to the Author

Reviewer #1: In this paper Brennan et al sought to evaluate the yield and linkage-to-care for diabetes and hypertension screening alongside a study assessing the use of rapid antigen tests for COVID-19 in taxi ranks in Johannesburg, South Africa. Overall, the paper is interesting and easy to follow. I do however have some remarks.

1. Throughout the manuscript and abstract elevated blood glucose is defined as fasting glucose >7 or random glucose >11.1 mmol/L, and elevated blood pressure as systolic >140 or diastolic >90 mmHg. However, the diagnostic criteria for both diabetes and hypertension include the cutoff-value i.e, glucose ≥7 and/or ≥11.1 mmol/L, and blood pressure ≥140 and/or ≥90 mmHg. Unless there is a strong rationale not to use the diagnostic cutoffs, I suggest reporting the results throughout the manuscript based on the diagnostic cutoffs.

2. Abstract, findings: The authors write that the prevalence of diabetes and hypertension was 7.1% and 27.9% respectively. This is not correct as a diagnosis of diabetes and hypertension require repeated elevated values. I suggest rephrasing the sentence.

3. Methods, first paragraph: Could you please provide more details how the blood pressure was measured? Was it measured after some rest, in seated, standing, or supine position? Were manual or automatic blood pressure meters used? Were previous diabetes and/or hypertension diagnoses self-reported?

4. Methods, outcomes: Could you please provide a reference to the waist circumference cutoffs indicative of the metabolic syndrome?

5. Methods, statistical analysis & Table 2: As I understand, the outcomes in the Poisson regression models were elevated blood glucose and elevated blood pressure, not diabetes and hypertension (se comment #3)? This also needs to be clarified in Table 2 as there are discrepancies in the table title and the column headings, and in the results and discussion sections. Could you also please add what statistical software that was used in the analyses?

6. Results, primary outcomes third paragraph: Typo – “in our cohort” is duplicated.

7. Results, primary outcomes fourth paragraph: Inconsistency regarding how many participants with elevated blood glucose that had no known previous diagnosis of diabetes who linked to care. In table 1, it says 18 participants but, in the text, and in Figure 1 there were 19 participants. Also, the text says it was 13 males and 5 females.

8. Results, predictors, first paragraph: The aRR of several of the reported predictors have wide 95% CI including 1, meaning that the effect of those variables are non-significant and therefore cannot be interpreted as predictors of increased risk of elevated blood glucose. Only the aRR regarding elevated blood pressure at enrollment and BMI ≥25 were significantly increased. Please only report significant findings as predictors of elevated blood pressure (as was done in the second paragraph).

9. Results, predictors, third paragraph: As I understand the results, no predictors of linkage-to-care were found, as none of the aRR for the mentioned variables in the paragraph and Table 3 were significantly increased or decreased. Please only report significant findings as predictors.

10. Discussion, first sentences in paragraphs one and two: Please see comment #3.

11. Discussion, sixth paragraph: Please rephrase and only report significant findings as predictors.

12. Discussion, last paragraph: Could the blood pressures have been elevated as the participants were also tested by potentially painful or unpleasant nasopharyngeal covid tests? If so, this might be added as a potential limitation.

13. Table 1: The table title states N=1168 but total N=1169.

14. References: #33 is not numbered and #34 seem to be missing in the reference list.

15. I couldn’t find the anonymized datasets at the OpenBU repository. Could you please provide a detailed link to the dataset?

Reviewer #2: This is a well written manuscript describing disease prevalence and awareness and linkage to care for diabetes and hypertension at a community-based COVID-19 testing venue in South Africa. The methods are well described and this manuscript fully meets the requirements set forth by PLOS ONE for publishing scientifically sound studies. I have a few suggestions for consideration about the implications of this work that the authors may consider.

1. Was there information collected on prior screening for diabetes or hypertension? This would be interesting to help understand the component causes of low rates of diagnosis (lack of screening vs screening that failed to identify these conditions).

2. Though still not adequate, as the authors note, diabetes and hypertension control among those aware of their diagnosis was much higher than seen in many other published studies, including those cited. What do the authors make of this finding? I wonder if disease control once someone is engaged with the health system is higher in this setting, or if simply those who self-report a prior diagnosis are also more likely to be engaged in care and adherent to treatment.

3. Linkage was quite low and while mentioned briefly in the discussion, I think this should be discussed further. Linkage was also much lower than other studies including older studies from a different context in South Africa (e.g. Govindasamy PLOS ONE 2013, among others). It would be worth discussing linkage in the present study in the context of other community-based hypertension and diabetes screening studies.

4. Further, regarding the low linkage rate, is there any other information about this particular context that may provide insight into the low rates of linkage to care (beyond the individual factors that are reported)? Distance from the taxi stand where recruitment occurred to the referral clinic? Distance from where participants lived if they were commuting from another part of the city? Availability of medications in the clinic? Reputation of the clinic in the community? Linkage to other clinics separate from the one to which they were referred? Further discussion of the potential factors contributing to low linkage to care could help readers contextualize this finding and be hypothesis-generating for potential interventions.

6. PLOS authors have the option to publish the peer review history of their article (what does this mean?). If published, this will include your full peer review and any attached files.

Reviewer #1: No

Reviewer #2: No

---

## [Author Response · Author response to Decision Letter 0]

26 Apr 2023

Reviewer #1: In this paper Brennan et al sought to evaluate the yield and linkage-to-care for diabetes and hypertension screening alongside a study assessing the use of rapid antigen tests for COVID-19 in taxi ranks in Johannesburg, South Africa. Overall, the paper is interesting and easy to follow. I do however have some remarks.

We thank the reviewer for their kind assessment

1. Throughout the manuscript and abstract elevated blood glucose is defined as fasting glucose >7 or random glucose >11.1 mmol/L, and elevated blood pressure as systolic >140 or diastolic >90 mmHg. However, the diagnostic criteria for both diabetes and hypertension include the cutoff-value i.e, glucose ≥7 and/or ≥11.1 mmol/L, and blood pressure ≥140 and/or ≥90 mmHg. Unless there is a strong rationale not to use the diagnostic cutoffs, I suggest reporting the results throughout the manuscript based on the diagnostic cutoffs.

We apologize for the oversight on our part. The cut-offs are set at the values you specified and should have read as fasting glucose >7 or random glucose >11.1 mmol/L, and elevated blood pressure as systolic >140 or diastolic >90 mmHg. We have changed this throughout the manuscript.

2. Abstract, findings: The authors write that the prevalence of diabetes and hypertension was 7.1% and 27.9% respectively. This is not correct as a diagnosis of diabetes and hypertension require repeated elevated values. I suggest rephrasing the sentence.

We thank the reviewer for their comment. We updated the results section in the abstract to match the results in the main text. It now reads as follows,

“1169 participants were enrolled and screened for diabetes and hypertension. Combining participants with a previous diagnosis of diabetes (n=23, 2%; 95% CI:1.3-2.9%) and those that had an elevated BG measurement (n=60, 5.2%; 95% CI:4.1-6.6%) at study enrollment, we estimated an overall indicative prevalence of diabetes of 7.1% (95% CI:5.7-8.7%). When combining those with known hypertension at study enrollment (n=124, 10.6%; 95% CI:8.9-12.5%) and those with elevated BP (n=202; 17.3%; 95% CI:15.2-19.5%), we arrive at an overall prevalence of hypertension of 27.9% (95% CI:25.4-30.1%). Only 31.7% of those with elevated BG and 16.0% of those with elevated BP linked-to-care.” 

3. Methods, first paragraph: Could you please provide more details how the blood pressure was measured? Was it measured after some rest, in seated, standing, or supine position? Were manual or automatic blood pressure meters used? Were previous diabetes and/or hypertension diagnoses self-reported?

The blood pressure measurement was taken while the patient was seated and with a manual blood pressure device with cuff. We have updated the sentence in the methods section on page 3 to state the following, 

“…and measured blood pressure (taken while the individual was seated via a once-off measurement with a manual blood pressure device with cuff)”.

We updated the methods section to reflect that previous hypertension and diabetes were self-reported. The statement in statistical analysis section of the paper on page 4 now reads as follows, 

“We modeled the outcome of diabetes as a function of age (18-29.9, 30-39.9, 40-49.9, 50-59.9, >60 years), sex, BMI (categorized as <25 vs. >25 kg/m2), smoking status (ever vs. never), elevated blood pressure at enrollment (diastolic >90 mmHg and systolic >140 mmHg), previous diabetes diagnosis at enrollment (self-reported) and previous hypertension diagnosis at enrollment (self-reported).”

4. Methods, outcomes: Could you please provide a reference to the waist circumference cutoffs indicative of the metabolic syndrome?

We have added the following references to the paper and have referenced them in the section where we define our outcomes in our methods (page 4).

The reference for our cut point of 90.0 cm for men has been added as citation #18: Kalk WJ, Joffe BI, Sumner AE. The waist circumference of risk in black South African men is lower than in men of European ancestry. Metab Syndr Relat Disord. 2011 Dec;9 (6):491-5. doi: 10.1089/met.2011.0063. Epub 2011 Aug 29. PMID: 21875336; PMCID: PMC3225062.

The reference for our cut point of 91.5 cm for females has been added as citation #19: Crowther NJ, Norris SA. The current waist circumference cut point used for the diagnosis of metabolic syndrome in sub-Saharan African women is not appropriate. PLoS One. 2012;7 (11):e48883. doi: 10.1371/journal.pone.0048883. Epub 2012 Nov 8. PMID: 23145009; PMCID: PMC3493601.

5. Methods, statistical analysis & Table 2: As I understand, the outcomes in the Poisson regression models were elevated blood glucose and elevated blood pressure, not diabetes and hypertension (se comment #3)? This also needs to be clarified in Table 2 as there are discrepancies in the table title and the column headings, and in the results and discussion sections. Could you also please add what statistical software that was used in the analyses?

We have updated the header in the Table 2 and updated the statistical analysis section on page 4 to read as follows, 

“Poisson regression to assess predictors of elevated blood glucose, elevated blood pressure, and linkage-to-care.” We also added the following to the end of the section, “All analyses were conducted using SAS v. 9.4.”

6. Results, primary outcomes third paragraph: Typo – “in our cohort” is duplicated.

The sentence on page 5 of the results now reads as follows, 

“Having a waist circumference indicative of metabolic syndrome (>90 cm for males and >91.5 cm for females) was rare in our cohort (0.7%; 95% CI:0.3-1.3%) (Table 1)”

7. Results, primary outcomes fourth paragraph: Inconsistency regarding how many participants with elevated blood glucose that had no known previous diagnosis of diabetes who linked to care. In table 1, it says 18 participants but, in the text, and in Figure 1 there were 19 participants. Also, the text says it was 13 males and 5 females.

We have updated the text, table and figure to reflect 13 males and 5 females linked to care. We updated the text in the results section on page 5 to read, 

“Amongst those participants with elevated blood glucose that had no known previous diagnosis of diabetes (n=60), 18 (n=30.0%; 95% CI:19.4-42.4%) linked to care within three weeks of study enrollment (Figure 1), while amongst those with elevated blood pressure and no known hypertension diagnosis (n=202), 33 (16.3%; 95% CI:11.7-21.9%) linked to care (Figure 2). Of those participants with elevated blood glucose, more males (n=13, 37.1%; 95% CI:22.5-53.9%) sought out PHC services than females (n=5; 20.0%; 95% CI:7.7-38.9%), but more females (n=13; 19.1%; 95% CI:11.1-29.8%) with elevated blood pressure linked to PHC services than males (n=20; 14.9%; 95% CI:9.6-21.7%) (Table 1).”

8. Results, predictors, first paragraph: The aRR of several of the reported predictors have wide 95% CI including 1, meaning that the effect of those variables are non-significant and therefore cannot be interpreted as predictors of increased risk of elevated blood glucose. Only the aRR regarding elevated blood pressure at enrollment and BMI ≥25 were significantly increased. Please only report significant findings as predictors of elevated blood pressure (as was done in the second paragraph).

We thank the reviewer for their comment. We have updated the language on page 6 in that paragraph to be less definitive, 

“The modified Poisson regression model we used to assess predictors of diabetes suggests that those >30 years of age, individuals with elevated blood pressure (aRR 1.96; 95% CI:1.15-3.33) and those with a BMI >25kg/m2 vs. <25kg/m2 (aRR 2.05; 95% CI:1.06-3.95) (Table 2). Our results, although imprecise, also suggest participants with a previous diagnosis of hypertension (adjusted risk ratio (aRR) 1.54; 95% CI:0.81-2.95), those with a previous diagnosis of diabetes (aRR 2.24; 95% CI:0.77-6.57), and those that had ever smoked vs never smokers (aRR 1.59; 95% CI:0.85, 2.97) at enrollment were at increased risk of elevated blood glucose.”

9. Results, predictors, third paragraph: As I understand the results, no predictors of linkage-to-care were found, as none of the aRR for the mentioned variables in the paragraph and Table 3 were significantly increased or decreased. Please only report significant findings as predictors.

We appreciate the reviewer’s feedback. We have updated the language on page 6 in that paragraph to be less definitive, 

“For the outcome of linkage-to-care, our results, although imprecise, suggest that those >40 years of age, participants with a previous diagnosis of hypertension (aRR 1.78; 95% CI:0.91-3.49) and individuals with a BMI >25kg/m2 vs. <25kg/m2 (aRR 1.98; 95% CI:0.83-4.73) were predictors of linkage-to-care (Table 3). Additionally, those with elevated blood pressure alone at enrollment were less likely to seek care than those with elevated blood sugar alone (aRR 0.68; 95% CI:0.31-1.51), while those with both elevated blood glucose and blood pressure were more likely to seek care than those with elevated blood glucose alone (aRR 1.23; 95% CI:0.46-3.29).“

10. Discussion, first sentences in paragraphs one and two: Please see comment #3.

We apologize but we are unclear on how comment #3 above relates to the first sentence in paragraph 1 of the discussion. Comment #3 is in reference to how blood pressure was measured and how previous diagnoses of hypertension and diabetes were reported. 

We assume the reviewer means the 3rd sentence in paragraph 1 and that we need to update our reference to hypertension and diabetes to elevated blood pressure and elevated blood glucose. We updated the following sentence in the first paragraph of the discussion on page 6, 

“Our study provides evidence of the feasibility of leveraging existing COVID-19 health screening infrastructure in South Africa to provide simple screening for elevated blood pressure and elevated blood glucose outside of health facilities at a high traffic taxi rank in Germiston.”

Same for the 1st sentence of the 2nd paragraph on page 6. We assume that the reviewer is referring to changing the language around diabetes diagnosis. We updated the following sentence in the second paragraph of the discussion, 

“We screened 1169 participants during an eight-week period and found an overall indicative prevalence of diabetes of 7.1% (combining previous diagnosis of diabetes (2%) and those that had an elevated blood glucose measurement (5.2%) at study enrollment).”

We made the same change in reference to hypertension diagnosis. We updated the following sentence in the third paragraph of the discussion on page 7, 

“Our overall hypertension prevalence was 27.9% (combining those with known hypertension at study enrollment (10.6%) and those with elevated BP (17.3%)), which is comparable to those previously reported out of South Africa (22-24).”

11. Discussion, sixth paragraph: Please rephrase and only report significant findings as predictors.

We thank the reviewer for their comment. We are clear in our language about the imprecise estimates in our discussion in reference to predictors of linkage to care. We state the following in paragraph 6 of the discussion on page 7, 

“Although our sample size of patients referred to care after an elevated blood glucose and/or blood pressure measurement was small (n=234) and we cannot draw strong conclusions, the predictors of linkage-to-care that we identified were comparable to what has been previously reported for chronic health conditions in this setting (20,30).”

12. Discussion, last paragraph: Could the blood pressures have been elevated as the participants were also tested by potentially painful or unpleasant nasopharyngeal covid tests? If so, this might be added as a potential limitation.

We thank the reviewer for their comment. Blood pressure measurements were taken prior to the administration of the nasopharyngeal swab for COVID-19 testing. We added the following to the first paragraph of the methods section to clarify study procedure’s, 

“All study relevant measures were recorded before nasopharyngeal swabs for COVID-19 testing were administered, eliminating the role these swabs could have had on elevating blood pressure.”

13. Table 1: The table title states N=1168 but total N=1169.

We updated the total in the subheading to N=1168 to match the title. Since we stratified the table by sex and data on sex was missing for one individual, we changed the numbers to reflect that.

14. References: #33 is not numbered and #34 seem to be missing in the reference list.

We apologize for the error. The references have been updated as follows:

#33 Gómez-Olivé, F. X., Ali, S. A., Made, F., Kyobutungi, C., Nonterah, E., Micklesfield, L., Alberts, M., Boua, R., Hazelhurst, S., Debpuur, C., Mashinya, F., Dikotope, S., Sorgho, H., Cook, I., Muthuri, S., Soo, C., Mukomana, F., Agongo, G., Wandabwa, C., Afolabi, S., … AWI-Gen and the H3Africa Consortium (2017). Regional and Sex Differences in the Prevalence and Awareness of Hypertension: An H3Africa AWI-Gen Study Across 6 Sites in Sub-Saharan Africa. Global Heart, 12 (2), 81–90. 

#34 Statistics South Africa. P0211 – Quarterly labour force survey (QLFS), 4th quarter 2022. 25 November 2008. Available at: https://www.statssa.gov.za/?page_id=1861&PPN=P0211&SCH=72944.

15. I couldn’t find the anonymized datasets at the OpenBU repository. Could you please provide a detailed link to the dataset?

You can locate the datasets at the following OpenBU link https://hdl.handle.net/2144/46066. We have updated the link in the paper.

Reviewer #2: This is a well written manuscript describing disease prevalence and awareness and linkage to care for diabetes and hypertension at a community-based COVID-19 testing venue in South Africa. The methods are well described and this manuscript fully meets the requirements set forth by PLOS ONE for publishing scientifically sound studies. I have a few suggestions for consideration about the implications of this work that the authors may consider.

1. Was there information collected on prior screening for diabetes or hypertension? This would be interesting to help understand the component causes of low rates of diagnosis (lack of screening vs screening that failed to identify these conditions).

We thank the reviewer for their comment and agree that it would be interesting to have that information. Unfortunately, we did not ask about previous screening, only about previously known diagnosis of hypertension and diabetes.

2. Though still not adequate, as the authors note, diabetes and hypertension control among those aware of their diagnosis was much higher than seen in many other published studies, including those cited. What do the authors make of this finding? I wonder if disease control once someone is engaged with the health system is higher in this setting, or if simply those who self-report a prior diagnosis are also more likely to be engaged in care and adherent to treatment.

We thank the reviewer for their comment. We updated the 5 paragraph of the discussion on page 7 to read as follows,

“We found 17.4% of those with a prior diabetes diagnosis and 26.0% of those with a prior diagnosis of hypertension at enrollment did not have their condition under control, the majority of which were female. Our estimates are lower than what has been reported previously in sub-Saharan Africa (uncontrolled diabetes ranging from 18.1% to 30.3% (20,30) and uncontrolled hypertension ranging from 50% to 93% (32-34)). The difference between our estimates and those previously reported could be that most of the previous estimates are from household surveys with a denominator comprised of everyone who was found in testing as part of the survey, while our denominator was comprised of individuals of people who know they have hypertension and/or diabetes. It also could be that disease control once someone is engaged with the health system is higher in this setting, or those who self-report a prior diagnosis are more likely to be engaged in care and adherent to treatment, even in the context of the COVID-19 pandemic.”

3. Linkage was quite low and while mentioned briefly in the discussion, I think this should be discussed further. Linkage was also much lower than other studies including older studies from a different context in South Africa (e.g. Govindasamy PLOS ONE 2013, among others). It would be worth discussing linkage in the present study in the context of other community-based hypertension and diabetes screening studies.

We thank the reviewer for their comment. We agree that linkage to care was low. We have added the following to paragraph 6 of the discussion on page 7,

“Linkage-to-care was poor in our study. It could be that moving testing for diabetes and hypertension into an individual’s commute makes it more difficult to link-to-care compared to being screened and diagnosed at a health facility where an individual could be enrolled in care for their condition and initiated onto treatment immediately, at the same clinic. A recent study assessing the intermediary cost-effectiveness of distributing HIV self-test kits and onward linkage to confirmatory testing and treatment services through 11 distribution models in South Africa reported that moving the location of HIV testing from the facility to the community helped close the gap in knowledge of HIV status but increased the gap in the next step, from diagnosis to engaging in care, as the health facility was still far away from the community and severe barriers to care continued to exist (36). Additionally, available evidence suggests that the COVID-19 pandemic severely impacted control and prevention programs for other conditions in South Africa, including diabetes and hypertension (37,38), due to higher demand for COVID-19 related health services, process changes including. screening of patients at the gate, longer than usual waiting times and line cutoffs, the need to book a visit in advance) and overall hesitancy of individuals to visit health facilities for any services during the pandemic.”

4. Further, regarding the low linkage rate, is there any other information about this particular context that may provide insight into the low rates of linkage to care (beyond the individual factors that are reported)? Distance from the taxi stand where recruitment occurred to the referral clinic? Distance from where participants lived if they were commuting from another part of the city? Availability of medications in the clinic? Reputation of the clinic in the community? Linkage to other clinics separate from the one to which they were referred? Further discussion of the potential factors contributing to low linkage to care could help readers contextualize this finding and be hypothesis-generating for potential interventions.

We agree that this information would be useful to have. However, when we referred patients to care, it was not to a specific clinic near the taxi rank, but rather we gave them a referral form with the details of the glucose and blood pressure measurements and then encouraged them to follow-up at their local clinic, which may have or may not have been close to the taxi rank. We think the paragraph added in response to comment #3 above can help add context.

---

## [Decision Letter · Decision Letter 1]

29 May 2023

PONE-D-22-33612R1Integration of point-of-care screening for type 2 diabetes mellitus and hypertension with COVID-19 rapid antigen screening in Johannesburg, South AfricaPLOS ONE

Dear Dr. Brennan,

Thank you for submitting your manuscript to PLOS ONE. After careful consideration, we feel that it has merit but does not fully meet PLOS ONE’s publication criteria as it currently stands. Therefore, we invite you to submit a revised version of the manuscript that addresses the points raised during the review process.

Please address the further comments presented by Reviewer 2. In addition, the phrase "*although imprecise most likely due to small sample size" *which is stated in the last paragraph of the results is better suited as a limitation.

We look forward to receiving your revised manuscript.

Kind regards,

Mobolanle Balogun

Academic Editor

PLOS ONE

Journal Requirements:

Reviewers' comments:

Reviewer's Responses to Questions

**Comments to the Author**

1. If the authors have adequately addressed your comments raised in a previous round of review and you feel that this manuscript is now acceptable for publication, you may indicate that here to bypass the “Comments to the Author” section, enter your conflict of interest statement in the “Confidential to Editor” section, and submit your "Accept" recommendation.

Reviewer #1: (No Response)

Reviewer #2: All comments have been addressed

2. Is the manuscript technically sound, and do the data support the conclusions?

Reviewer #1: Yes

Reviewer #2: Yes

3. Has the statistical analysis been performed appropriately and rigorously? 

Reviewer #1: Yes

Reviewer #2: Yes

4. Have the authors made all data underlying the findings in their manuscript fully available?

Reviewer #1: Yes

Reviewer #2: Yes

5. Is the manuscript presented in an intelligible fashion and written in standard English?

Reviewer #1: Yes

Reviewer #2: Yes

6. Review Comments to the Author

Reviewer #1: The manuscript has improved and been clarified in the revised version. I just have some minor remarks regarding the results section about Predictors of diabetes, hypertension and linkage-to-care:

1. As stated in the methods section and tables the predictors are about elevated blood glucose, elevated blood pressure and linkage-to-care and I suggest rephrasing the subheading accordingly.

2. The last bit of the first sentence in the first paragraph seem to be missing.

3. As the 95% CIs are wide and include 1 indicating no effect, I suggest toning down the results described in the last sentence of the third paragraph in a similar way as the author have done with other non-significant findings. For example, an aRR of 0.68 with 95% CI 0.31–1.51 really means that individuals could be anywhere from 69% less likely to 51% more likely to seek care.

Reviewer #2: (No Response)

7. PLOS authors have the option to publish the peer review history of their article (what does this mean?). If published, this will include your full peer review and any attached files.

Reviewer #1: No

Reviewer #2: No

---

## [Author Response · Author response to Decision Letter 1]

3 Jun 2023

Reviewer #1: The manuscript has improved and been clarified in the revised version. I just have some minor remarks regarding the results section about Predictors of diabetes, hypertension and linkage-to-care:

We thank the reviewer for reviewing our manuscript a second time and providing useful feedback.

1. As stated in the methods section and tables the predictors are about elevated blood glucose, elevated blood pressure and linkage-to-care and I suggest rephrasing the subheading accordingly.

We thank the reviewer for their comment. We have made changes throughout the manuscript to make it clear that our study is assessing elevated blood glucose, blood pressure and linkage-to-care and not diabetes or hypertension.

2. The last bit of the first sentence in the first paragraph seem to be missing.

The sentence now reads, “The modified Poisson regression model we used to assess predictors of diabetes elevated blood glucose suggests that those >30 years of age, individuals with elevated blood pressure at enrollment (adjusted risk ratio (aRR) 1.96; 95% CI:1.15-3.33) and those with a BMI >25kg/m2 vs. <25kg/m2 (aRR 2.05; 95% CI:1.06-3.95) were at increased risk of elevated blood glucose (Table 2).

3. As the 95% CIs are wide and include 1 indicating no effect, I suggest toning down the results described in the last sentence of the third paragraph in a similar way as the author have done with other non-significant findings. For example, an aRR of 0.68 with 95% CI 0.31–1.51 really means that individuals could be anywhere from 69% less likely to 51% more likely to seek care.

We thank the reviewer for their comment. The sentence now reads, “We also found, although imprecise, that those with only elevated blood pressure at enrollment were less likely to seek care than those with only elevated blood sugar (aRR 0.68; 95% CI:0.31-1.51), while those with both elevated blood glucose and blood pressure were more likely to seek care compared to those with only elevated blood glucose (aRR 1.23; 95% CI:0.46-3.29).

---

## [Editor Report · Decision Letter 2]

13 Jun 2023

Integration of point-of-care screening for type 2 diabetes mellitus and hypertension with COVID-19 rapid antigen screening in Johannesburg, South Africa

PONE-D-22-33612R2

Dear Dr. Brennan,

We’re pleased to inform you that your manuscript has been judged scientifically suitable for publication and will be formally accepted for publication once it meets all outstanding technical requirements.

Kind regards,

Mobolanle Balogun

Academic Editor

PLOS ONE
---

## [Editor Report · Acceptance letter]

29 Jun 2023

PONE-D-22-33612R2 

Integration of point-of-care screening for type 2 diabetes mellitus and hypertension with COVID-19 rapid antigen screening in Johannesburg, South Africa 

Dear Dr. Brennan:

I'm pleased to inform you that your manuscript has been deemed suitable for publication in PLOS ONE. Congratulations! Your manuscript is now with our production department. 

Kind regards, 

on behalf of

Dr. Mobolanle Balogun 

Academic Editor

PLOS ONE